# New Markers in Monitoring the Reactivation of Hepatitis B Virus Infection in Immunocompromised Hosts

**DOI:** 10.3390/v11090783

**Published:** 2019-08-25

**Authors:** Valentina Svicher, Romina Salpini, Vincenzo Malagnino, Lorenzo Piermatteo, Mohammad Alkhatib, Carlotta Cerva, Loredana Sarmati

**Affiliations:** 1Department of Experimental Medicine and Surgery, University of Rome Tor Vergata, 00133 Rome, Italy; 2Clinic of Infectious Diseases, Department of System Medicine, University of Rome Tor Vergata, 00133 Rome, Italy

**Keywords:** hepatitis B infection, HBV, HBV markers, HBsAg, HBV core related antigen, HBcrAg, HBV-DNA, anti-HBcAb, HBV-RNA, immunocompromised host

## Abstract

Hepatitis B virus (HBV) persistence is at the basis of HBV reactivation as a consequence of chemotherapy and immunosuppressive treatments. The identification of early viral replication indicators and markers of effective HBV immunological control would be useful in monitoring patients who are at risk of potential viral reactivation during the course of immunosuppressive treatment. Currently, international guidelines have shared some criteria to identify patients with a low, medium or high risk of HBV reactivation; however, permanently placing a patient in a definitive category is not always easy. More often, patients move from one category to another during the course of their immunosuppressive treatment; therefore, in many cases, there are no precise indicators or tools for monitoring possible reactivation and establishing the duration and suspension of antiviral prophylaxis. Historically, the sequence of HBV antigens and antibodies and HBV DNA levels has been used to evaluate the different stages of the acute and chronic phases of an HBV infection. In the last few years, new biomarkers, such as anti-HBs and anti-HBc titres, HBV core-related antigen (HBcrAg), ultra-sensitive HBsAg evaluation and HBV RNA, have been used in patients with an HBV infection to evaluate their diagnostic and prognostic potential. The aim of this review is to evaluate the published results on the use of new infection markers in the diagnosis and monitoring of HBV reactivation over the course of immunosuppressive treatments. Moreover, the importance of viral genotypic studies was emphasized, given the diagnostic and therapeutic implications of the mutational profiles of HBsAg during the HBV reactivation phase.

## 1. Introduction

Patients with a hepatitis B virus (HBV) infection remain at risk of serious consequences, such as chronic hepatitis (CHB) or hepatocellular carcinoma (HCC), conditions that are more frequent in some regions of the world due to the high risk of infection acquisition in childhood and to the presence of some particularly aggressive viral genotypes [1]. Currently, anti-HBV treatments inhibit HBV replication, increasing the survival rates and quality of living in CHB patients; however, they cannot completely eliminate HBV from hepatic cells. Therefore, to date, there is no definitive cure for HBV infection.

HBV persistence is a common condition not only in patients with CHB but also in patients who have completely recovered from natural infection, and it is at the basis of HBV reactivation in cancer patients and in other immunocompromised subjects as a consequence of chemotherapy and immunosuppressive treatments.

HBV reactivation (HBV-R) in immunocompromised hosts can be associated with extremely serious clinical consequences. A retrospective Japanese study reported 20% fulminant hepatic failure with a subsequent 100% mortality rate in HBV-R in haematological patients, which resulted in a higher risk of mortality associated with fulminant HBV hepatitis than otherwise healthy people with acute HBV infection [2].

Recently, the American Gastroenterological Association [3] classified immunosuppressive treatment as low (<1% of cases), moderate (1–10% of cases), or high risk (>10% of cases) of HBV-R based on the knowledge from a systematic review. Based on the risk attributable to the drugs, the condition of the patient and the duration of treatment, indications of screening, prevention and prophylaxis have been established for the different categories of patients at risk of HBV-R. However, it is not always easy to identify the real risk in patients who switch from one immunosuppressive therapy to another, patients who are treated for a long time, or haematological and transplant patients with a high grade of immunosuppression. In this kind of subject, the grade of HBV-R risk and the length of immunosuppression are not always clearly quantifiable; therefore, many unclear areas remain regarding the duration of monitoring for HBV-R, prophylaxis or pre-emptive therapy. In these categories of patients, the identification of early viral replication markers or the best indicators of HBV immunological control would be useful in monitoring patients at risk of potential viral reactivation or to identify the right time for the suspension of antiviral prophylaxis.

A series of viral markers, such as HBsAg, HBeAg, anti-HBc, anti-HBe and anti-HBs, has been historically used to evaluate the natural course and to establish the phases of HBV infection. Moreover, the detection of HBV-DNA levels has become a fundamental practice for establishing the extent of viral replication and guiding the start of therapy. In particular, HBV-DNA level monitoring is recommended in immune-compromised CHB patients receiving antivirals to monitor for reactivation before the end of antiviral treatment and periodically thereafter. In patients with resolved HBV infection (anti-HBc positive anti HBsAg negative), there is no shared opinion supporting HBV-DNA monitoring for reactivation; however, watchful monitoring may also be a reasonable choice (i.e.; in the course of rituximab treatment) [4,5].

Recently, new biomarkers, such as titres of anti-HBs and anti-HBc, HBV core-related antigen (HBcrAg), ultra-sensitive HBsAg evaluation, and HBV-RNA, have been used in patients with HBV infection to evaluate their diagnostic and prognostic potential [6,7].

The aim of this review is to evaluate the existing literature on the use of new HBV markers in the assessment of the immunocompromised patient at risk for HBV-R to understand the clinical areas in which they could be wisely used and discuss the issues that need additional research results. In the final paragraph, a brief discussion of the characteristics of the viral strains isolated during HBV-R in patients with immunodeficiency is reported. The relevance of this topic is related to the changes in the HBV genome frequently reported in these cases and often associated with manifestations of viral immune escape and drug resistance, which can complicate the diagnosis and treatment processes.

## 2. Literature Search

The literature review was based on a search strategy for English-language literature in PubMed and the Cochrane Library. The research was performed by the authors, who independently reviewed all articles for study inclusion. The search terms used either alone or in combination were ‘hepatitis B virus,’ ‘HBV,’ ‘new hepatitis B viral biomarkers’, ‘immunocompromised host’, ‘HBsAg’, ‘HBsAg titer’, ‘ultra-sensitivity HBsAg titer’, ‘HBV core related antigen’, ‘HBcrAg’, ‘HBV-DNA’, ‘anti-HBcAb’, ‘HBV-RNA’, and ‘hepatitis B virus reactivation’. Studies evaluating non-immunocompromised patients were excluded.

A total of 10 studies containing the use of new HBV viral biomarkers in the evaluation of HBV infection reactivation in immunocompromised hosts were included for review. In all the selected studies, the reactivation of HBV infection (defined as an abrupt appearance/rise in HBV-DNA levels and a reverse seroconversion to an HBsAg-positive status (reappearance of HBsAg)) concerned immunosuppressed people.

Of the 10 studies evaluated, six were on anti-HBsAg titre, two studies were on anti-HBcAb titre, one on HBcrAg and one on ultra-sensitive HBsAg (Table 1). For the relevance and novelty of the results, an abstract presented at the “52nd Annual Meeting of the Italian Association for the Study of the Liver” on the theme of the review was also considered.

## 3. Role of New markers in Monitoring Patients at Risk of HBV Reactivation

A series of viral biomarkers of HBV have been recently correlated with the clinical evolution and response to therapy in HBV-infected patients. For some of these, recent research data suggest a possible practical use, while others are still being studied for possible future applications.

Below is the list of the new markers of HBV infection with their most significant meanings resulting from the published studies.

Anti-HBs titres: A reduction or a disappearance of anti-HBs titres has been correlated with a major risk of HBV relapse.

Anti-HBc titres: High titres of HBcAb have been correlated with HBeAg clearance [18], the efficacy of antiviral therapy, and the absence of HBV relapse to treatment suspension [19].

HBcrAg: HBcrAg is considered to be a marker of intrahepatic HBV cccDNA. A decrease in HBcrAg has been related to the loss of HBeAg and HBsAg and the safe discontinuation of HBV treatment [20].

HBV-RNA: Serum pre-genomic RNA transcribed from cccDNA has a strong correlation with intrahepatic cccDNA and is considered to be an indirect marker of reservoir size [21].

Ultra-sensitive HBsAg: HBsAg quantification (qHBsAg) levels are correlated with intra-hepatic covalently closed circular (ccc) DNA [22].

Below, the results of studies that have speculated applications for the above markers in immunocompromised patients are reported (Table 1).

### 3.1. Anti-HBs Titre

Although the quantification of anti-HBs has been available for several years, the clinical relevance of anti-HBs titres in the context of immunosuppression has only recently been investigated. In particular, the role of anti-HBs titres in predicting HBV-R was first analysed in a study of 260 patients with resolved HBV infection receiving rituximab-containing chemotherapy [9]. In this study, HBV-R occurred in 30.2% of patients after a median of 23 weeks of rituximab treatment [9]. An undetectable anti-HBs titre (<10 mIU/mL) at baseline was the only significant risk factor correlated with the increased risk of HBV-R (hazard ratio (HR): 3.51 (95% confidence interval (CI): 1.37–8.98); *p* = 0.009). Indeed, patients negative for anti-HBs at baseline had a significantly higher 2-year cumulative rate of HBV-R compared to patients positive for anti-HBs (68.3% vs 34.4%; *p* = 0.012). In a subsequent study, Cho Y. and collaborators [10] showed that in a group of 108 patients with B-cell lymphoma, those with an anti-HBsAg titre <100 IU/mL had a risk of reactivating HBV equal to 8.3% and 17.3% after 6 and 24 months of rituximab treatment, respectively.

In line with these results, another study focused on 336 patients with resolved HBV infection undergoing kidney transplantation, and the patients were stratified according to the use of rituximab for desensitization [12]. The authors found that standard-dose rituximab was significantly associated with a steeper decline in anti-HBs titres compared to reduced-dose rituximab. Furthermore, standard-dose rituximab (HR (95% CI): 10.6 (2.52–44.60), *p* = 0.001) and anti-HBs  < 100 IU/L at transplantation (HR (95% CI): 9.06 (1.11–74.30); *p* =  0.04) were independent risk factors for HBV-R. In the context of solid tumours, a recent study showed that an undetectable anti-HBs titre and a dexamethasone dose >1.0 mg/day were high risk factors of HBV-R (OR (95% CI): 5.94 (1.15–30.6), *p* = 0.03, and 8.69 (1.27–58.8), *p* = 0.02, respectively) [13].

Similar results were obtained from two large prospective studies from Japan [11] and China [14] on the risk of the reactivation of hepatitis B in a large number of patients treated with immunosuppressive drugs for a rheumatic disease. In the Japanese study, which enrolled 1042 patients over 2 years, low (<100 mIU/mL) or no (<10 mIU/mL) anti-HBsAg titres were one of the risk factors (together with age >69 years) for HBV-R. In the Chinese study, 380 patients were prospectively evaluated from 2013 to 2017 during antirheumatic therapy with biological agents. The study showed an incidence rate of HBV-R of 4.7/100 persons a year, 2.5/100 persons a year and 0/100 persons a year for the groups of patients with a negative anti-HBsAg titre (<10 UI mL), a low anti-HBsAg titre (10–100 IU/mL), and a high anti-HBsAg titre (>100 IU/mL), respectively.

These results were confirmed also by a recent study showing that an anti-HBs titer < 100 mIU/mL at baseline is the only virological marker correlated with a higher risk of developing HBV-R (55.6% of pts with anti-HBs < 100 mIU/mL vs 0% with anti-HBs > 100 mIU/mL experienced HBV-R, *p* = 0.046) [23].

Overall, these findings support the idea that the progressive weakening of humoural responses can play an important role in driving HBV-R. A low/undetectable anti-HBs titre can thus be useful in identifying patients at higher HBV-R risk who could benefit from vaccination to elevate the antibody titres.

### 3.2. Anti-HBc Titres

Beyond anti-HBs, anti-HBc titres can play a role in identifying patients at a higher HBV-R risk who can be candidates for antiviral prophylaxis. In this regard, a recent study has analysed liver biopsies from 100 transplant donors positive for anti-HBc with standard serologic testing [24]. cccDNA was detected in 52% of samples [24]. The authors found that the median anti-HBc titres were significantly higher in cccDNA-positive than negative liver tissues (17.0 (7.0–39.2) vs. 5.7 (3.6–9.7) cut-off index (COI), *p* = 0.007) [24]. Interestingly, by multivariate analysis, an anti-HBc IgG titre >4.4 COI was correlated with cccDNA detection (OR: 8.516, *p* = 0.009). Thus, such a threshold can be indicative of cccDNA persistence in the liver of patients with a resolved HBV infection. It is conceivable that such cccDNA can drive the production of viral antigens (albeit at minimal levels) capable of stimulating anti-HBc production.

In only two studies was the anti-HBc titre correlated with reactivation in the immunocompromised host, and it is always associated with the presence or absence of antibodies to HBsAg. Yang and colleagues [16] focused on 197 patients with lymphoma and resolved HBV infection. In this study, HBV-R occurred in 12.2% of patients with an incidence of 11.6/100 person-years. Interestingly, the authors found that patients with an anti-HBc titre ≥ 6.41mIU/mL and an anti-HBs titre < 56.48 mIU/mL were characterized by the highest risk of HBV-R (HR (95% CI): 17.29 (3.92–76.30), *p* < 0.001) and by an inferior overall survival (HR (95% CI): 2.41 (1.15–5.05), *p* = 0.02) [16]. According to these results, the authors suggest that baseline anti-HBc/anti-HBs levels can help identify patients at high risk of HBV-R who should receive antiviral prophylaxis. In this regard, a Japanese study conducted on 77 patients with lymphoma and signs of resolved hepatitis shows that the combination of anti-HBc and anti-HBs levels may be useful for predicting the development and timing of chemotherapy-induced hepatitis [15]. Unexpectedly, the authors demonstrated that anti-HBsAg-negative patients with high anti-HBc titres (>10 signal to cut off ratio (S/CO)) prior to chemotherapy experienced a significantly higher rate of HBV reactivation than patients with both low anti-HBc (<10 S/CO) and high anti-HBs levels (>28 mIU/mL) who did not completely reactivate.

### 3.3. HBcrAg

A unique recent study focused on the role of HBcrAg in predicting HBV-R [17]. By analysing 124 oncohaematological patients with serologically resolved HBV infection, Seto and colleagues showed that HBcrAg positivity at baseline was significantly associated with HBV-R (HR (95% CI): 2.94 (1.43–6.07), *p* = 0.004). In particular, HBcrAg-positive patients were characterized by a significantly higher HBV-R rate than HBcrAg-negative patients (71.8 vs. 31%, *p* = 0.002). Furthermore, in a subgroup of patients receiving rituximab, positivity to HBcrAg and negativity to anti-HBs were independent factors associated with a higher HBV-R risk (HR (95% CI): 3.65 (1.35–9.86), *p* = 0.011 and 2.84 (1.10–7.37), *p* = 0.032, respectively).

Overall, these results support the role of HBcrAg in identifying patients with transcriptionally active cccDNA who can be more prone to HBV-R, thus benefitting from antiviral prophylaxis.

### 3.4. Serum HBV-RNA

As mentioned in the previous paragraph, another intriguing biomarker is represented by serum HBV-RNA. This biomarker measures the number of virions containing pre-genomic RNA that are released from infected hepatocytes. Interestingly, recent studies have shown that RNA-containing virions can be the prominent type of virion during nucleoside analogue (NUC) treatment since pre-genomic RNA is not converted into DNA by HBV reverse transcriptase [25,26]. The only demonstration of a correlation between HBV-RNA elevation and immunosuppressive drugs is an in vitro study in which the exposure of HepG2 cells to epirubicin determined elevated levels of HBV-DNA and pre-genomic HBV-RNA [27].

Until now, the role of serum HBV-RNA in the setting of human immunosuppression-driven HBV-R has not been investigated, even if it is possible to argue that this biomarker could help in identifying a transcriptionally active cccDNA during the administration of antiviral prophylaxis, thus helping in optimizing the duration of antiviral prophylaxis.

### 3.5. Ultra-Sensitive HBsAg

The lower limit of detection (LLOD) of the currently available assays for HBsAg detection is 5 mIU/mL. Thus far, more sensitive assays for HBsAg detection and quantification (LLOD: 0.5 mIU/mL) have been developed. These assays could be useful in patients with a resolved HBV infection to identify minimal cccDNA transcriptional activity and, in turn, minimal HBV replication. Shinkai and co-authors [8] compared three ultra-sensitive HBsAg assays to study serum samples from 120 patients with haematological malignancy receiving systemic chemotherapy; in particular, in 13 subjects with demonstrated HBV-R by HBV-DNA elevation. The authors show how a semi-automated immune complex transfer chemiluminescence enzyme technique (ICT-CLEIA) was able to detect the presence of HBsAg on 12 of 13 patients (in two cases, even before HBV-DNA elevation) with a sensitivity comparable to the evaluation of HBV-DNA. Despite these interesting results, there are limited applications of these assays in the field of immunocompromised patients. A recent observation suggests that the on-monitoring analysis of ultra-sensitive HBsAg might be another risk factor for HBV-R [23]. Nevertheless, further investigation is needed to unravel this topic which could help in identifying patients prone to being HBV-R candidates for antiviral prophylaxis.

## 4. Future Perspectives on the Use of Biomarkers for Predicting HBV-R

Overall, the findings support the use of innovative biomarkers in identifying patients who are more prone to developing HBV-R and who are candidates to receive antiviral prophylaxis. Nevertheless, further issues should be addressed. In particular, the role of innovative biomarkers has been mainly investigated in oncohaematological patients with a resolved HBV infection at a higher risk of HBV-R. Further studies are needed to unravel the role of biomarkers in patients with an intermediate or a low risk of HBV-R. This is critical for the tailored management of preventive strategies for patients at risk of HBV-R.

The current guidelines recommend antiviral prophylaxis administered for 12–18 months after stopping the administration of immunosuppressive drugs in patients at a high risk of HBV-R with HBeAg-negative CHB or with resolved HBV infection [28]. Nevertheless, the duration of antiviral prophylaxis is still under debate, since several cases of HBV-R have been described after prophylaxis suspension, particularly in the setting of profound immune suppression [29,30,31,32,33,34]. The role of biomarkers in identifying patients with or without a transcriptionally active cccDNA deserves further investigation to identify patients who should continue or interrupt antiviral prophylaxis.

Finally, another setting in which the role of innovative biomarkers should be investigated is represented by HIV co-infection. Interestingly, a recent study has highlighted the issue of HBV reactivation in anti-HBc positive HBsAg-negative patients switching to an antiretroviral regimen not including drugs with anti-HBV activity [35]. Studies addressing the role of biomarkers in detecting minimal HBV replication can help to identify patients who should continue or should safely suspend tenofovir.

## 5. Detection of Mutational Profiles in HBsAg by Genotypic Testing

The biomarkers described so far can be useful when starting an immunosuppressive treatment to identify patients with a higher risk of HBV-R, whereas, in patients experiencing HBV-R, the characterization of mutational profiles in HBsAg can be useful. Indeed, several studies have highlighted a high degree of genetic variability of HBsAg in patients experiencing immune-suppression-driven HBV-R [31,32,36,37,38,39,40]. In particular, a previous study identified specific genetic signatures of HBsAg that were significantly correlated with the status of HBV-R [32]. Notably, the majority of these mutations reside in the major hydrophilic HBsAg region (MHR), the main B cell HBsAg epitope target of neutralizing antibodies, and are known to hamper HBsAg recognition from antibodies. Among them, D144E has been implicated in immune escape and/or diminished affinity for monoclonal antibodies [41]. This mutation has also been described in a case of the corticosteroid-induced reactivation of latent hepatitis B virus infection in an HIV-positive patient [37]. Similarly, an enrichment of mutations at HBsAg position 145 (G145A/R) was detected in patients with HBV-R. Mutations at this position are known not only to reduce the HBsAg affinity for neutralizing antibodies but also to abrogate the production of neutralizing antibodies in a murine model [42]. Overall, the clustering of mutations associated with HBV-R in HBsAg immune-active regions suggests that these mutations may contribute to the re-uptake of viral replication during the progressive weakening of the immune system.

Recent studies have shown that the presence of immune-escape mutations at the start of anti-HBV treatment can hamper the subsequent achievement of HBsAg loss [43,44]. Thus, such mutations can hinder the return to an HBsAg-negative status during NUC treatment in HBV-reactivated patients with a previously resolved HBV infection.

Furthermore, recent studies have shown that a substantial proportion (approximately 20% in European studies) of patients who were HBsAg-negative and anti-HBc-positive at screening remained HBsAg-negative despite the reactivation of viral replication and high levels of serum HBV DNA [32,36,45]. Again, this can be ascribed to the genetic characteristics of HBsAg observed in patients with HBV-R. In particular, a previous study has highlighted an enrichment of *N*-linked glycosylation sites in HBsAg in patients remaining HBsAg negative despite viral reactivation [32]. *N*-linked glycosylation sites play a critical role in the viral evasion from humoural responses since the glycan shield can mask the B-cell epitope target of neutralizing antibodies [46,47]. In this regard, a recent study has shown that the N-linked glycosylation sites detected in HBV-reactivated patients profoundly affect HBsAg recognition by antibodies in vitro [32]. These findings reinforce the role of immune-escape mutations in immune-suppression-driven HBV-R. At the same time, this highlights the importance of considering serum HBV-DNA or innovative HBsAg assays targeting HBsAg epitopes outside the major hydrophilic region (MHR) in the diagnosis of HBV-R.

Finally, due to the peculiar organization of the HBV genome, the gene encoding HBsAg overlaps with the gene encoding reverse transcriptase (RT). This implies that some immune-escape HBsAg mutations can introduce mutations in HBV RT, which can have implications for the efficacy of lamivudine prophylaxis. Indeed, previous studies have demonstrated that these RT mutations can restore the viral replication capacity impaired by the classical lamivudine resistance mutations M204I/V [48,49]. This suggests that the high degree of HBsAg variability observed in patients with HBV-R may provide a genetic backbone that may contribute to the emergence of drug-resistance mutations. This concept also has implications in the setting of antiviral prophylaxis, supporting the use of more potent anti-HBV drugs to prevent HBV-R, particularly in patients with chronic HBeAg-negative infection.

## 6. Conclusions

In conclusion, a series of new biomarkers have been correlated with the different phases of HBV infection; however, as mentioned above, their applications in the diagnosis and monitoring of HBV-R during immunosuppressive treatments are insufficient to establish their correct positioning in the diagnosis of HBV-R.

To date, it seems likely that anti-HBs and anti-HBc titre assays could be the most readily applicable assays. Their use in clinical practice could reasonably provide useful information about the level of specific protective responses during immunosuppressive treatments and stimulate possible considerations regarding the use of the HBV vaccine with the purpose of enhancing the immune response. The ultra-sensitive HBsAg levels assay could replace HBV-DNA detection; however, we are still far from the routine use of tests for the assessment of the HBV reservoir (HBcrAg, HBV-RNA).

Finally, it seems useful to reinforce the importance of resistance mutation studies on HBV isolates from patients experiencing a reactivation of the infection, given the relevant implications that they have in the diagnostic and therapeutic fields.

## Figures and Tables

**Table 1 viruses-11-00783-t001:** Summary of studies led in patients with resolved hepatitis B virus (HBV) infection addressing the role of HBV biomarkers in predicting HBV reactivation.

Advantages of New HBV Markers Tests	Reference	No. of Patients	Clinical Setting	% of HBV-R *	Association of Biomarker with Risk of HBV-R
	**HBsAg Titer**				
HBsAg detection at lower limit (0.5 mIU/mL) predictive of HBV-R	Skinkai et al. Liver Int. 2017 [8]	120 (13 with HBV-RNA detectable)	Oncohematological HBV-resolved patients	100%	Architect HBsAg-QT (detection limit:50 mIU/mL), HBsAg-HQ (5 mIU/mL) and ICT-CLEIA (0.5 mIU/mL) were used to test stored samples. ICT-CLEIA was detected in all 12 patients (100%) with a sensitivity comparable to HBV-DNA
	**Anti-HBs Titer**				
Low anti-HBsAg levels (<100 UI/mL) and major risk of HBV-R	Seto et al. 2014 [9]	260	Oncohematological patients receiving rituximab	30.2%	Anti-HBs negativity at baseline: HR (95%CI):3.51(1.37–8.98), *p* = 0.009
	Cho Y et al. 2016 [10]	108	B-cell lymphoma receiving rituximab not antiviral prophylaxis	11.6%	Anti-HBs < 100 IU/mL HBV-R rates at 6, 12, 36, and 48 months after chemotherapy as high as 8.3, 17.3, 21.1, and 25.7%
	Fukuda et al. 2016 [11]	1042	Rheumatic diseases	3.3%	Anti-HBs < 100 IU/mL OR (95% CI): 2.8 (1.3 to 6.8)
	Lee et al. 2018 [12]	366	Kidney-transplanted patients using rituximab for desensitization	2.5%	Anti-HBs < 100 IU/L at transplantation: HR (95% CI): 9.06 (1.11–74.3), *p* = 0.04 standard-dose rituximab at transplantation: HR (95% CI): 10.60 (2.52–44.60), *p* = 0.001
	Kotake et al. 2018 [13]	243	Patients with solid tumors	2.1%	Anti-HBs negativity at baseline: OR (95% CI):5.94 (1.15–30.6), *p* = 0.03 dexamethasone > 1.0 mg/day at baseline: OR (95% CI): 8.69 (1.27–58.8), *p* = 0.02
	Tien et al. 2018 [14]	380	Rheumatic diseases treated with biologic therapy	4.4%	Anti-HBs > 100 IU/mL at baseline Rate of HBV-R person year 0/100 Anti-HBs 10–100 IU/mL at baseline Rate of HBV-R person year 2.5/100 Anti-HBs < 10 IU/mL at baseline Rate of HBV-R person year 4.7/100
	**Anti HBc Titer**				
Combination of anti-HBc/anti-HBs low levels identify high risk of HBV-R	Matsubara et al. 2017 [15]	77	Lymphoma patients	12.9%	Anti-HBcAb < 10 (S/CO) at baseline OR(95% CI) 0.11 (0.013–0.665), *p* = 0.016 Anti-HBcAb < 28 (mUI/mL) at baseline OR(95% CI) 10.5 (1.749–105.993), *p* = 0.009
	Yang et al. 2018 [16]	197	Oncohematological patients	12.2%	AntiHBs < 56.48 mIU/mL + anti-HBc ≥ 6.41IU/mL at baseline: HR (95% CI):17.29 (3.9–76.3), *p* < 0.001
	**HBcrAg Presence**				
HBcrAg positivity associated with HBV-R	Seto et al. 2016 [17]	124	Oncohematological patients	25%	HBcrAg positivity at baseline: HR (95% CI): 2.94 (1.43–6.07), *p* = 0.004
		62	Oncohematological patients receiving rituximab	29%	HBcrAg positivity at baseline: HR(95% CI):3.65(1.35–9.86), *p* = 0.011 anti-HBs negativity at baseline: HR (95% CI): 2.84 (1.10–7.37), p = 0.032

* HBV-R = HBV reactivation; ICT-CLEIA: immune complex transfer chemiluminescence enzyme technique.

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
