# Peer review of "New Markers in Monitoring the Reactivation of Hepatitis B Virus Infection in Immunocompromised Hosts"

_viruses, 2019, doi:10.3390/v11090783_

Round 1

Reviewer 1 Report

This review paper summarized the recently published papers on new markers for monitoring the reactivation of HBV infection in immunocompromised patients. 

suggest to replace the table 1 into comparison of new HBV markers (advantages, disadvantages, and limitations) for predicting HBV reactivation instead of listing of references etc.  Many typoes and spacing and inconsistency of reference formating should be corrected.

Author Response

Reviewer 1

1.suggest to replace the table 1 into comparison of new HBV markers (advantages, disadvantages, and limitations) for predicting HBV reactivation instead of listing of references etc.  Many typoes and spacing and inconsistency of reference formating should be corrected.

Answer - In agreement with  reviewer's request, a new column with the advantages offered by the use of new HBV markers in the immunocompromised patient, has been included in table 1. For clarity of reading, the references column, supporting the evidence of the results, has been left.

Reviewer 2 Report

In this paper, Svicher et al. reviewed recent papers showing the usefulness of HBV serum markers for the prediction of HBV reactivation. The reactivation of HBV after chemotherapy or immunosuppression therapy had been difficult to predict, but recent several papers reported especially the importance of HBsAb and HBcAb titers. The manuscript is written well and covers the issues regarding the theme. I have some minor comments as below.

In Table 1, there are some typos such as ‘Skinai’, ‘HBVNA’, and ‘Anti-HBscAb’. Please unify the terms ‘HBV reactivation’ and ‘HBV-R’. I think the section 3 (New biomarkers of HBV infection) is not essential, because its most part is duplicate of next section. Please check the style of author names in Reference 19.

Author Response

Manuscript ID: viruses-571277

Type of manuscript: Review

Title: New markers in monitoring the reactivation of hepatitis B virus infection in immunocompromised hosts

Reviewer 2

1.In Table 1, there are some typos such as ‘Skinai’, ‘HBVNA’, and ‘Anti-HBscAb’.

Answer - We agree with the reviewer, table 1 has been extensively reviewed and the typos corrected

2. Please unify the terms ‘HBV reactivation’ and ‘HBV-R’.

Answer - In agreement with the reviewer, HBV reactivation was replaced by the acronym HBV-R in the table and in the text.

3. I think the section 3 (New biomarkers of HBV infection) is not essential, because its most part is duplicate of next section.

Answer - According  with the reviewer, the paragraph 3 has been removed, part of the text has been reported in the next paragraph. The paper’s paragraphs have been renumbered as a result of the change, and now the text as a whole is made up of 6 paragraphs

4. Please check the style of author names in Reference 19

Answer - The style of the names in reference 19 has been revised and corrected